artificial intelligence/human-computer interaction

answer selection, attention mechanism, bidirectional LSTM, convolutional neural network, Siamese network

**Author for correspondence:**
Yuan Wei
e-mail: weiyuan0315@sina.com

# Double attention recurrent convolution neural network for answer selection

## Ganchao Bao, Yuan Wei, Xin Sun and Hongli Zhang

School of Mechatronic Engineering and Automation, Shanghai University, Shanghai 200444, People's Republic of China

YW, 0000-0002-9776-3615

Answer selection is one of the key steps in many question answering (QA) applications. In this paper, a new deep model with two kinds of attention is proposed for answer selection: the double attention recurrent convolution neural network (DARCNN). Double attention means self-attention and cross-attention. The design inspiration of this model came from the transformer in the domain of machine translation. Self-attention can directly calculate dependencies between words regardless of the distance. However, self-attention ignores the distinction between its surrounding words and other words. Thus, we design a decay self-attention that prioritizes local words in a sentence. In addition, cross-attention is established to achieve interaction between question and candidate answer. With the outputs of self-attention and decay self-attention, we can get two kinds of interactive information via cross-attention. Finally, using the feature vectors of the question and answer, elementwise multiplication is used to combine with them and multilayer perceptron is used to predict the matching score. Experimental results on four QA datasets containing Chinese and English show that DARCNN performs better than other answer selection models, thereby demonstrating the effectiveness of self-attention, decay self-attention and cross-attention in answer selection tasks.

## 1. Introduction

Question answering (QA) is an important and challenging task in the field of natural language processing (NLP). It has a wide range of applications in the fields of intelligent online customer service and intelligent assistants. Answer selection is one of the key steps in many QA applications and can be expressed as, given a question and an answer candidate pool $\{a_1, a_2 \ldots, a_s\}$, our goal is to pick the answer that matches the question from the pool of candidate answers. The main challenge of this task is that the correct answer may not have the vocabulary mentioned in the question. Therefore, questions and answers may only be semantically related.

In recent years, deep learning has achieved significant successful processing tasks in various natural languages, such as semantic analysis [1], machine translation [2], text abstract [3] and other intelligent domains, such as automatic speech recognition [4,5], intelligent fault diagnosis [6–9] and smart factory [10–12]. It has also achieved good performance in answer selection [13–16]. Compared with traditional models [17,18], deep learning has several advantages. For example, it can automatically extract complex features, while traditional models require hand-designed features. Deep learning can capture semantic features, while traditional models only use surface lexical features.

The deep model of the convolutional neural network (CNN) [13] is often used in answer selection. However, CNN can only analyse local semantic information due to the limitation of filter size and cannot capture global semantic information. The recurrent neural network (RNN) and its associated variants, including long–short-term memory (LSTM) [19] and gated recursive units (GRU) [20], can capture current and previous information from forward and backward. However, if the length of the sequence is too long, it is still difficult for the RNN model to learn remote dependencies. RNN may not capture the long-term dependency information between the words in the sequence.

In this paper, to improve the accuracy of answer selection task, a new deep model, double attention recurrent convolution neural network (DARCNN), is proposed based on double attention. The bidirectional LSTM (BiLSTM), self-attention, decay self-attention, cross-attention and CNN are combined as the deep model to extract global features, local features and interactive information of the question and candidate answers, and make the semantic modelling of questions and candidate answers in multiple dimensions. Better feature vectors of question and answers can thus be obtained, and the matching score can be predicted by MLP more accurately.

The contributions of DARCNN are briefly outlined as follows:

(1) DARCNN uses two attention mechanisms: self-attention and cross-attention. The internal structure of the two attention mechanisms is the same, but the inputs are different, resulting in completely different functions. Self-attention can be used for global semantic modelling of questions and answers and is not limited by long-range distance in the sequence. To get more local information in a sentence, we propose a variant of self-attention, named decay self-attention, along with a decay matrix. Cross-attention can also describe the interaction of questions and answers, and allows questions and answers to generate their own weight of attention based on the other. At the same time, cross-attention can also capture dependencies between potentially matching question and answer pairs, which can provide additional information for text relevance for answer selection.

(2) In DARCNN, BiLSTM and CNN are also critical. BiLSTM is capable of contextual semantic modelling forward and backward, outputting semantic representation vector with certain word order. CNN complements the lack of semantic extraction of attention on local information, especially in adjacent words. In this study, the model uses a CNN block with three different size filers to extract local semantic information with multigranularity.

(3) Experimental results show that DARCNN performs better than many other networks when analysing the NLPCC DBQA, WikiQA, TrecQA and ANTIQUE datasets. Additionally, the effectiveness of every component of this model, including self-attention, decay self-attention and cross-attention, in the tasks of answer selection are analysed.

# 2. Related works

In the domain of deep learning, Yu *et al.* [21] proposed a convolutional bigram graph model to choose the right answer. Severyn & Moschitti [22] used CNN with dense layers to capture the interaction between question and candidate answers using tree kernels [23]. Wang & Nyberg [24] combined a stacked BiLSTM to learn a common representation vector of question and the candidate answer.

Recently, various forms of attention mechanisms have been applied to answer selection. Tan *et al.* [25] used the attentive BiLSTM that produced important weighting on pooling based on the relevance between the question and answer. Dos Santos *et al.* [26] proposed a two-way attention mechanism based on learning metrics for the similarity between questions and candidate answers. Wang *et al.* [27] proposed a new approach to integrating attention into and within a GRU. In RNN, gated attention explores semantic relations within sentences and made remarkable progress in natural language inference. BiMPM [28] matched sentences with multiple granularity features from multiple views. Inter-weighted alignment network (IWAN) [29] extracted features from a word alignment matrix using self-attention. Based on the existing extraction strategy, a new parametric self-attention

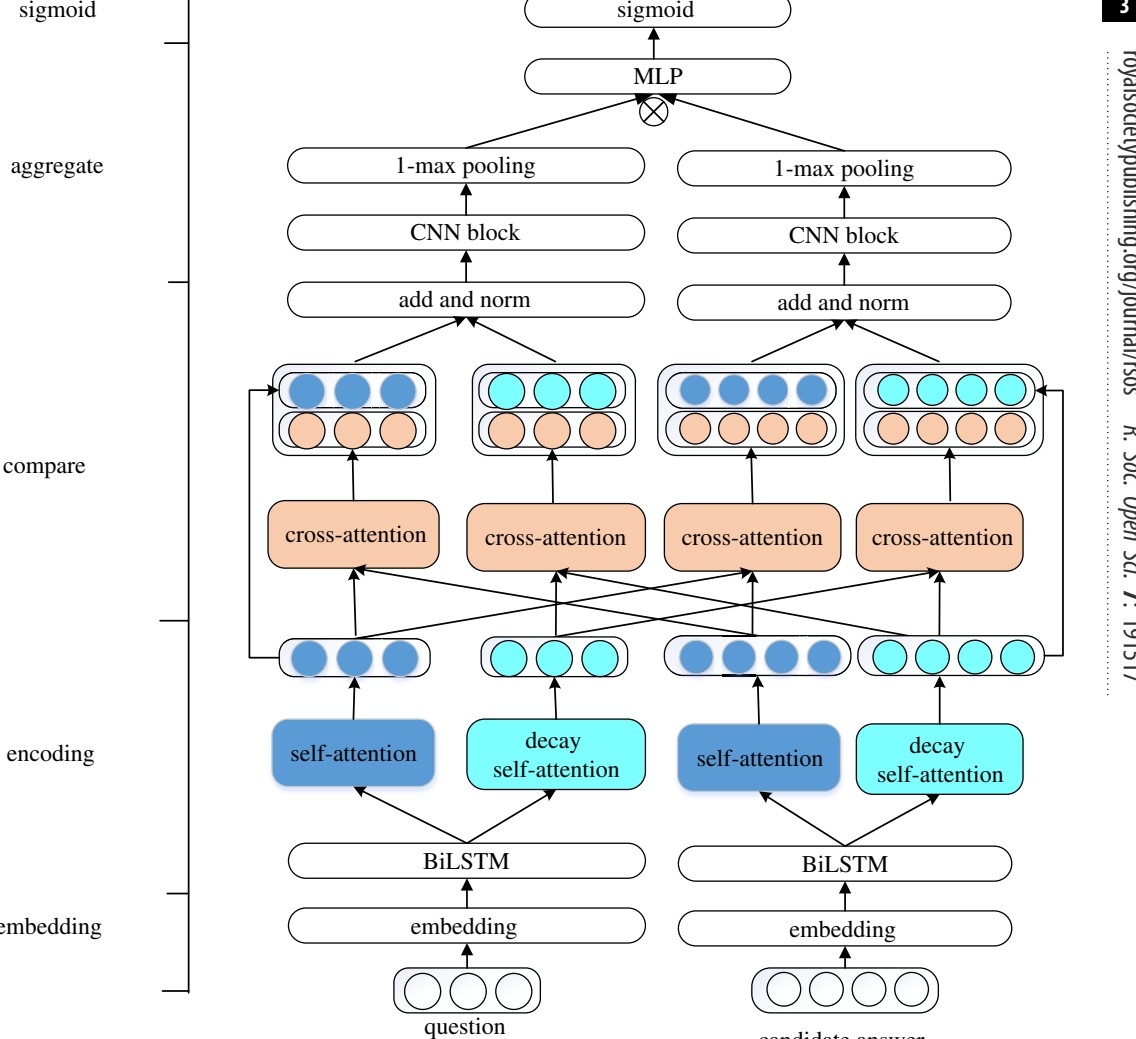

**Figure 1.** Architecture of the Siamese network with DARCNN.

fixed attention feature was designed. Thus, more attention mechanisms, such as internal attention and self-attention, are introduced with LSTM to extract interpretable sentence embedding.

Unlike these studies, we improve the existing self-attention and propose adding a decay mask on self-attention, called decay self-attention, to catch more dependency relations between surrounding words. This is important because surrounding words and local features may be more important than distant words and global features in answer selection. Our model uses self-attention and decay self-attention at the same time to get feature information from different perspectives. Then, cross-attention yields more interactive information between the question and candidate answer. Experimental analysis shows that the decay mask did have a positive effect on the model, and our model performed well in answer selection.

# 3. DARCNN model and methods

The DARCNN model is based on Siamese architecture [30], as shown in figure 1. A pretrained 300-dimensional word vector is used to embed text. BiLSTM can get contextual semantics in forward and backward text order. Self-attention allows text to focus on the dependency of the word on other words in the current time step to obtain global semantic information, while decay self-attention will pay more attention to the surrounding words. Cross-attention allows questions and answers to determine each answer's word-level attention weight. Then, multilayer CNN blocks create semantic representation vectors of the final question and answer. These semantic representation vectors are then merged by elementwise multiplication, and a matching score is generated by a multilayer perceptron (MLP) with the sigmoid function.

## 3.1. Siamese architecture

Siamese architecture is a successful framework for text matching that has a symmetrical component to extract high-level features from two input channels. Those channels share parameters and map inputs to the vectors with the same dimension. Then, we can merge two vectors and calculate matching scores. Where $f_{qu}$ and $f_{an}$ are the feature vectors of question and candidate answer, $\sigma$ represents the sigmoid function, $W_2$, $W_1$, $b_1^T$, $b_2^T$ represent the weight parameters and $\odot$ represents the dot-product, the fusion and matching process can be described by the following equation:

$$s = \sigma(W_2\text{ReLU}(W_1(f_{qu} \odot f_{an}) + b_1^T) + b_2^T). \tag{3.1}$$

The binary cross-entropy loss is used as a loss function of DARCNN model, where $y_i$ represents the label of 0 or 1 and $s$ is the output of the sigmoid function in the following equation:

$$L = -\sum_{i=1}^{N} [y_i \log(s_i) + (1 - y_i) \log(1 - s_i)]. \tag{3.2}$$

## 3.2. BiLSTM

LSTM was originally proposed by Hochreiter & Schmidhuber [19], and can mitigate gradient disappearance in an RNN. Because LSTM uses the adaptive gate mechanism, the gate can selectively pass information through a sigmoid neural layer and elementwise multiplication. Each element of the vector output by the sigmoid layer is a ratio between 0 and 1, representing how much corresponding information is passed. The LSTM has input, forget and output gates, which determine how much the LSTM maintains its previous memory and extracts current information. Given an input sequence $X = \{x(1), x(2), \ldots, x(n)\}$, $n$ is the length of the sentence, and $i_t$, $f_t$ and $o_t$ represent the input gate, forget gate and output gate, respectively. Where the parameters $\{W_i, U_i, b_i\}$, $\{W_f, U_f, b_f\}$ and $\{W_o, U_o, b_o\}$ are the weight matrices of the input gate, forget gate and output gate, respectively; $C_t$ represents the current cell state; $W_c$, $U_c$, $b_c$ represent the parameters of new memory content $\tilde{C}_t$; and $x(t)$ represents the input of the time $t$, the hidden vector $h(t)$ can be updated as follows:

$$i_t = \sigma(W_i x(t) + U_i h(t-1) + b_i), \tag{3.3}$$
$$f_t = \sigma(W_f x(t) + U_f h(t-1) + b_f), \tag{3.4}$$
$$o_t = \sigma(W_o x(t) + U_o h(t-1) + b_o), \tag{3.5}$$
$$\tilde{C}_t = \tanh(W_c x(t) + U_c h(t-1) + b_c), \tag{3.6}$$
$$C_t = i_t \odot \tilde{C}_t + f_t \odot C_{t-1}, \tag{3.7}$$

and
$$h_t = o_t \odot \tanh(C_t). \tag{3.8}$$

The bidirectional LSTM structure is used to obtain the context information in the text, as shown in figure 2. The disadvantage of LSTM is that it cannot use context information from future tokens. The BiLSTM generates two separate output vector sequences by processing the sequence in both directions with previous and future contexts: one processes the input sequence in the forward direction, and the other processes the input sequence in the backward direction. The output of each time step is a concatenation of the output vectors in both directions.

## 3.3. Double attention

As shown in figure 3, attention is calculated using a scaled dot-product of attention in the self-attention layer of our model. The model can determine attention weights of a single sequence to calculate its feature vector. In this way, it can capture the association between each word and other words in the sequence. The scaled dot-product of attention has inputs, namely, the three matrices $Q$ (Query), $K$ (Key) and $V$ (Value), which all come from the same input $X$. We can get $Q$, $K$, $V$ by multiplying $X$ with a matrix. First, we have to calculate the dot-product between $Q$ and $K$. Then, the result is divided by a scale $\sqrt{d_k}$ to prevent it from being too large. Then, the softmax function is used to normalize the result to a probability distribution and then multiplied by the matrix $V$ to get a new contextualized representation matrix. Where $W_i^Q \in \mathbb{R}^{d_{model} \times d_k}$, $W_i^K \in \mathbb{R}^{d_{model} \times d_k}$ and $W_i^V \in \mathbb{R}^{d_{model} \times d_v}$ are

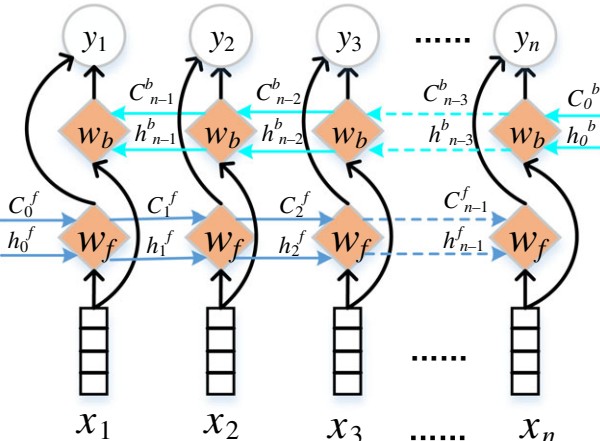

**Figure 2.** Calculation schematic of BiLSTM, where $W_f$ and $W_b$ represent the calculated operation of LSTM at each time step forward and backward, respectively.

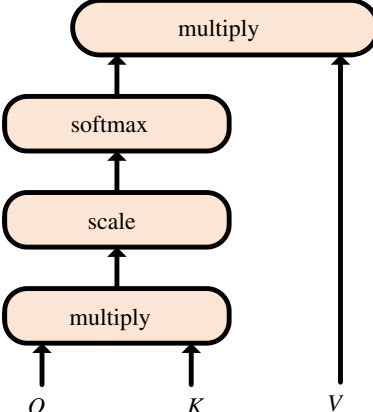

**Figure 3.** Calculation schematic of the scaled dot-product of attention.

the weight matrices of linear transformation, $\sqrt{d_k}$ is the dimension of the Query and Key vector, $\sigma$ is the softmax function, this operation can be described by the following equation:

$$Q = XW_i^Q, \quad K = XW_i^K, \quad V = XW_i^V, \tag{3.9}$$

$$\sigma(w) = \text{softmax}\left(\frac{Q*K^T}{\sqrt{d_k}}\right) \tag{3.10}$$

and
$$\text{Att} = \sigma(w)*V. \tag{3.11}$$

On the self-attention layer, multiple self-attentions are stacked to form the multihead attention, as shown in figure 4. Defining $h$ as the number of heads, each head learns features in different representation spaces so that the DARCNN model can extract more granular text feature.

First, Query, Key and Value are determined by linear transformation, and then we calculate $h$ times of scaled dot-product attention. Then, consolidating the results of $h$ times, the result of multihead attention after another linear transformation can be obtained. This instruction allows the model to learn relevant information in subspaces of different linear transformations. Given that the input of the self-attention layer is a matrix of $x = \{x_1, x_2, \ldots, x_n\}$, $n$ is the length of the sequence. In our self-attention layer, $Q$, $K$ and $V$ are obtained separately by $X$ multiplying by a weight matrix. For each $x_i$, the self-attention layer is calculated to compare with other vectors in the sequence and obtain the attention weight of $x_i$ to adjust the value of $x_i$. $\text{Att}_h$ is the output of each head attention, $W_i^O \in \mathbb{R}^{hd_v \times d_{\text{model}}}$ is parameter of

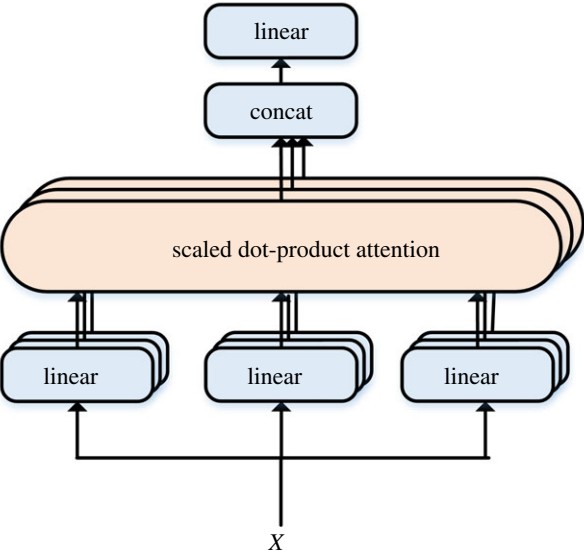

**Figure 4.** Calculation schematic of multihead attention.

|   | 0 | −1 | −2 | ... | 1−n |
|---|---|---|---|---|---|
| 1 | 0 | −1 | −2 | ... | 1−n |
| 2 | −1 | 0 | −1 | ... | 2−n |
| 3 | −2 | −1 | 0 | ... | 2−n |
| ⋮ | ... | ... | ... | ... | ... |
| n | 1−n | 2−n | 3−n | ... | 0 |
|   | 1 | 2 | 3 | ... | n |

**Figure 5.** Decay matrix of the decay self-attention.

linear transformation, as shown in the following equation:

$$\text{MultiAtt} = [\text{Att}_1, \dots, \text{Att}_h]W_i^O. \tag{3.12}$$

On this basis, we also used a variation of self-attention, called decay self-attention. The author of the transformer [31] proposed a masked multihead attention to prevent subsequent information from leaking into the translation process. Inspired by this, we propose a decay mask for multihead attention to allow our model to pay more attention to surrounding words. Based on equation (3.11), we add a decay matrix to the attention weight $\sigma(w)$, where $M_{\text{decay}} \in \mathbb{R}^{n \times n}$ is the decay matrix and $\alpha$ is the parameter of decay mask, as shown in the following equation:

$$\text{decayAtt} = (\sigma(w) + \alpha M_{\text{decay}}) * V. \tag{3.13}$$

We designed the decay matrix with this idea: attention weight attenuates as distance from the current word increases. As shown in figure 5, the value of $(i, j)$ in the decay matrix is $-|i - j|$, representing the decay degree between $i$th word and $j$th word. This value is multiplied by parameter $\alpha$ and added to the attention weight. Then, the attention weight decreases as distance increases. Thus, the model pays more attention to surrounding words.

Such a structure may seem to have the same function as CNN, focusing on local features. However, the difference between decay self-attention and CNN is that CNN only extracts local features within a fixed window size, while decay self-attention considers all words in a sentence at the same time by assigning unequal attention weights to pay more attention to local features and be smoothed.

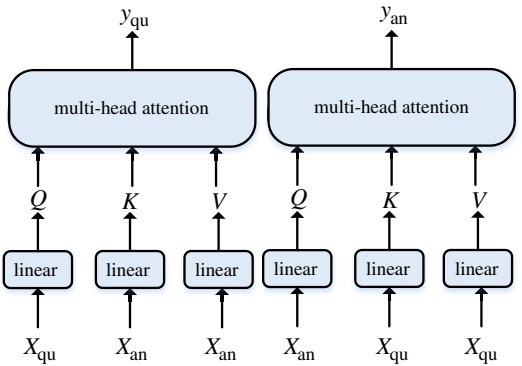

**Figure 6.** Calculation schematic of a question and answer in the cross-attention layer.

We combine the decay self-attention into a multihead attention form in the same way, as in equation (3.12). We use normal self-attention and decay self-attention, which can be used to obtain two representations of global and local features. Both of these features are put into the cross-attention layer.

Cross-attention has the same internal structure as self-attention but uses different inputs and a different function. As shown in figure 6, cross-attention also has three inputs, namely, $Q$ (Query), $K$ (Key) and $V$ (Value), but $Q$, $K$ and $V$ do not come from the same input. Defining the representation vector of question as $X_{qu} = \{w_1, w_2, \ldots, w_n\}$, the representation vector of the candidate answer as $X_{an} = \{w_1, w_2, \ldots, w_m\}$, where $n$, $m$ represents the length of the question and answer. Then, in the branch network of the question, the three inputs of the cross-attention layer are $\{X_{an}, X_{qu}, X_{qu}\}$, and in the branch network of the candidate answer, the three inputs of the cross-attention layer are $\{X_{qu}, X_{an}, X_{an}\}$. Where $W_i^{Q1}, W_i^{Q2} \in R^{d_{\text{model}} \times d_k}$, $W_i^{K1}, W_i^{K2} \in \mathbb{R}^{d_{\text{model}} \times d_k}$ and $W_i^{V1}, W_i^{V2} \in \mathbb{R}^{d_{\text{model}} \times d_v}$ are the weight matrices of linear transformation. $Q$, $K$, $V$ of question and answer is calculated as follows:

$$Q_{qu} = X_{qu} W_i^{Q1}, \quad Q_{an} = X_{an} W_i^{Q2}, \tag{3.14}$$

$$K_{qu} = X_{an} W_i^{K1}, \quad K_{an} = X_{qu} W_i^{K2} \tag{3.15}$$

and

$$V_{qu} = X_{an} W_i^{V1}, \quad V_{an} = X_{qu} W_i^{V2}. \tag{3.16}$$

When getting the $K$, $Q$, $V$ of a question and answer, the next calculation is the same as that for self-attention. In this way, the question and candidate answer implement the interaction of semantic information, which determines each other's representation by cross-attention.

The parameters $q_s$, $q_{ds}$, $a_s$ and $a_{ds}$ are represented by the output of self-attention and decay-attention for a question and candidate answer. Then, we can perform the above cross-attention calculation with these values to obtain interactive information between sentences. Calculating in pairs, we can determine $\tilde{q}_s$, $\tilde{q}_{ds}$, $\tilde{a}_s$, $\tilde{a}_{ds}$, which are represented by the output of cross-attention, as shown in the following equations:

$$\tilde{q}_s = \text{attention}(Q_{q_s}, K_{a_s}, V_{a_s}), \tag{3.17}$$

$$\tilde{q}_{ds} = \text{attention}(Q_{q_{ds}}, K_{a_{ds}}, V_{a_{ds}}), \tag{3.18}$$

$$\tilde{a}_s = \text{attention}(Q_{a_s}, K_{q_s}, V_{q_s}) \tag{3.19}$$

and

$$\tilde{a}_{ds} = \text{attention}(Q_{a_{sd}}, K_{q_{ds}}, V_{q_{ds}}). \tag{3.20}$$

Then, we concatenate two vectors to represent the final output: $[q_s, \tilde{q}_s]$, $[q_{ds}, \tilde{q}_{ds}]$, $[a_s, \tilde{a}_s]$ and $[a_{ds}, \tilde{a}_{ds}]$, as shown in figure 1. These parameters are combined on the dimension, so that the size of the vector becomes $n \times 2d$ from $n \times d$. The outputs of cross-attention have more semantic information, including interactive information between sentences. Finally, we respectively add outputs of the question and outputs of candidate answer, normalizing them to prevent values from becoming too large.

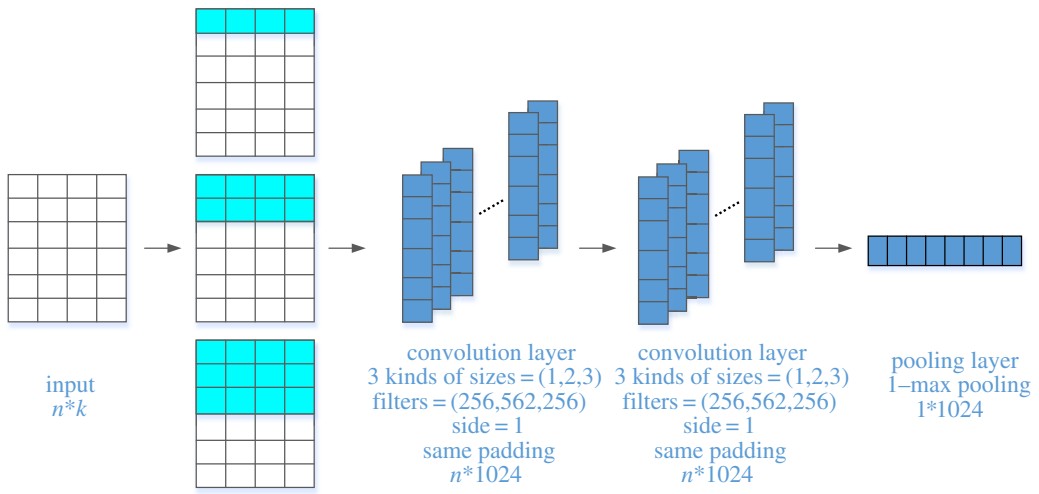

**Figure 7.** Calculation schematic of the CNN block.

**Table 1.** Statistics of datasets used for answer selection.

| dataset | train questions | valid questions | test questions | candidates per question | answer length in tokens | language |
|---|---|---|---|---|---|---|
| NLPCC DBQA | 8768 | — | 5997 | 20.6 | 38.4 | Chinese |
| WikiQA | 873 | 126 | 243 | 9.8 | 25.2 | English |
| TrecQA | 1162 | 68 | 65 | 38.4 | 30.3 | English |
| ANTIQUE | 2426 | — | 200 | 11.3 | 47.7 | English |

## 3.4. CNN block

Finally, the DARCNN model uses the CNN block to extract local semantic information of different granularity and obtain the representative vector of the candidate answer and question, as shown in figure 7. We define a sentence $X = \{x_1, x_2, \ldots, x_n\}$, where $n$ represents the length of the sentence. In the CNN block, the input matrix size is $n \times k$. Each CNN block is composed of a one-dimensional convolution of 1024 filters: 256 filters of $1 \times k$, 562 filters of $2 \times k$ and 256 filters of $3 \times k$. The one-dimensional convolution's step size is 1, and padding is used to maintain its same shape. Then, we obtain 1024 vectors of $n \times 1$ and combine them into $n \times 1024$ matrix to place into the next convolution layer. Our model uses different filters with sizes 1, 2 and 3 in a one-dimensional convolution operation to extract local information of different granularities, especially the semantic information of surrounding words. After passing the last convolution layer, we obtain a matrix of $n \times 1024$ and process it with 1-max pooling to obtain the final $1 \times 1024$ representation vector of the question or the candidate answer.

# 4. Experiments

## 4.1. Dataset

To prove the validity of the proposed method, experiments are performed on three datasets: NLPCC-2016 DBQA, WikiQA and TrecQA. The statistics of the datasets are listed in table 1.

## 4.2. Evaluation metrics

The standard experimental indicators of answer selection are the mean reciprocal rank (MRR) and mean average precision (MAP). Where $Q$ represents a set of questions, and $\text{rank}_i$ is the ranking position of the

first correct candidate answer of the $i$th question, the MRR can be calculated as follows:

$$\text{MRR} = \frac{1}{Q}\sum_{i=1}^{Q}\frac{1}{\text{rank}_i}. \tag{4.1}$$

However, MAP focuses on the ranks of all correct candidate answers. If the correct candidate answers for a question $q \in Q$ is $\{d_1, d_2, \ldots, d_{mj}\}$, and where $R_{jk}$ is the set of ranked retrieval results from the top result until you get to the answer $d_k$, then the MAP can be calculated with the following equation:

$$\text{MAP} = \frac{1}{Q}\sum_{j=1}^{|Q|}\frac{1}{m_j}\sum_{k=1}^{|m_j|}\text{Precision}(R_{jk}). \tag{4.2}$$

## 4.3. Experiment set-up

The best model for different datasets uses slightly different hyperparameters. All datasets use 300-dimension pretrained word embedding to obtain vectors. For the English dataset, the word vectors trained by the glove model on 6 billion words from Wikipedia are used. The Chinese dataset is different from the English dataset and must be separated first via jieba segmentation. The word vectors trained by word2vec model on Baidu Encyclopaedia data are used. Random initialization is used for unregistered words, and the length of the input depends on the maximum length of the question and answer. In NLPCC DBQA and ANTIQUE, the question length $L_q = 60$, and the candidate answer length $L_a = 120$. In TrecQA, the question length $L_q = 56$, and the candidate answer length $L_a = 200$. In WikiQA, the question length $L_q = 48$, and the candidate answer length $L_a = 200$. The number of hidden layer units of BiLSTM is 150 in each direction. The output dimensions of self-attention and cross-attention are consistent with the input dimensions. The head number is $h = 4$. Finally, one-dimensional convolution with multiple filters of sizes 1, 2 and 3 is used. The size of the hidden layer for MLP is 1024, and the final activate function is sigmoid. The rest of the activate functions are ReLU, the dropout rate is 0.5, and the batch size is set to 32. All models are optimized using the Adam algorithm, and the initial learning rate is 0.0001 and gradually decays to 0.00005.

## 4.4. Result

Table 2 shows the experimental results for the four datasets. The 12 models for comparison are used, and several are reimplemented. However, due to parameter adjustment, the accuracy of the redesigned model is generally lower than the accuracy proposed in the original paper. Whether using a Chinese or English dataset, the performance of DARCNN exceeds that of most other models. Compared with BERT, DARCNN is 1–2% points lower across the four datasets, but the advantage of DARCNN is that its model capacity is much smaller than BERT's.

# 5. Discussion

## 5.1. Result analysis

The baseline methods CNN and BiLSTM are compared in table 2. Using the four datasets, CNN's baseline approach yields better performance than BiLSTM's because the text length in these four datasets is relatively short, and CNN can capture local multigranularity semantic information well. Thus, CNN is more suitable for short text matching of a question and answer. The combination of CNN and BiLSTM can also improve accuracy. As BiLSTM can extract global semantic features in forward and backward order, it mitigates the shortcomings of CNN. In the DARCNN model, two attention mechanisms are used. Self-attention and decay self-attention assign an attention weight for each word based on a comparison of the current word and other words. The model will then extract important information from the text. Cross-attention describes the interaction of questions and candidate answers, and the attention weights of each word are assigned by comparing the questions and answers. The results show that the DARCNN model yields better performance after adding two kinds of attention mechanisms.

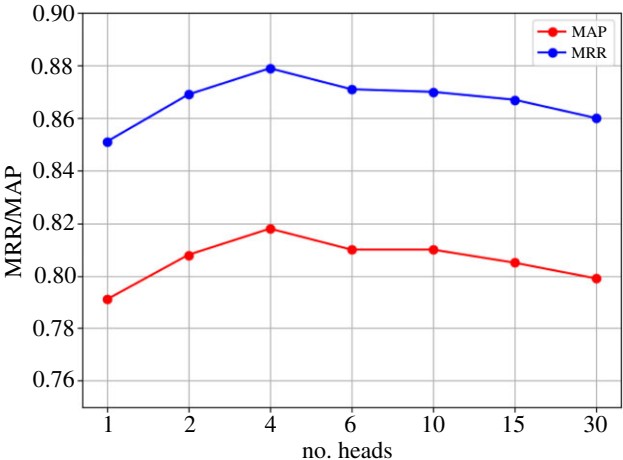

**Figure 8.** Result of a different number of heads in self-attention and cross-attention on TrecQA dataset.

**Table 2.** Result of different models on NLPCC DBQA, WikiQA, TrecQA and ANTIQUE datasets.

| model | NLPCC DBQA | | WikiQA | | TrecQA | | ANTIQUE | |
|---|---|---|---|---|---|---|---|---|
| | MAP | MRR | MAP | MRR | MAP | MRR | MAP | MRR |
| CNN | 0.729 | 0.735 | 0.6204 | 0.6365 | 0.661 | 0.742 | 0.161 | 0.396 |
| BiLSTM | 0.684 | 0.684 | 0.6174 | 0.6310 | 0.636 | 0.715 | 0.155 | 0.381 |
| CNN + BiLSTM | 0.748 | 0.750 | 0.6560 | 0.6737 | 0.678 | 0.752 | 0.174 | 0.424 |
| MP-CNN [32] | 0.771 | 0.772 | 0.670 | 0.679 | 0.709 | 0.788 | — | — |
| ABCNN [33] | 0.815 | 0.816 | 0.691 | 0.686 | 0.711 | 0.801 | — | — |
| IABRNN [27] | 0.828 | 0.828 | 0.684 | 0.691 | 0.728 | 0.819 | — | — |
| AP-BiLSTM [26] | 0.833 | 0.834 | 0.671 | 0.684 | 0.713 | 0.803 | — | — |
| MPCNN + NCE [34] | 0.836 | 0.837 | 0.701 | 0.718 | 0.783 | 0.859 | — | — |
| BiMPM [28] | 0.834 | 0.834 | 0.718 | 0.731 | 0.802 | 0.875 | — | — |
| MS-LSTM [16] | 0.852 | 0.853 | 0.711 | 0.724 | 0.800 | 0.877 | — | — |
| BERT [35] | — | — | 0.753 | 0.770 | 0.877 | 0.927 | 0.377 | 0.797 |
| DARCNN | 0.859 | 0.860 | 0.734 | 0.750 | 0.818 | 0.879 | 0.367 | 0.771 |

In the self-attention and cross-attention layers, multiple attentions that are similar are stacked to form multihead attention. The number of heads $h$ represents the calculated number of stacked attentions. A sensitivity study is thus designed for the hyperparameter $h$ on the TrecQA dataset. The input to the self-attention and cross-attention layers is 300 dimensions; thus, $h = 1, 2, 4, 6, 10, 15, 30$. Results are then compared for the MAP and MRR indices. The model under these different parameters is then trained, and the epoch is set to 50. We save the model and test it using test data at every epoch to obtain the highest MAP and MRR indices of 50 epochs in each experimental group. As is shown in figure 8, the model performs best in the self-attention and cross-attention layers when $h = 4$. In theory, more heads yield more attention that the model pays to the different sequence contents. However, when the number of heads is too large, performance is not improved because when there are too many heads, the dimension of the subspace is too small, and the contained information is insufficient.

In the CNN layer, the CNN block is a one-dimensional convolution combination of different filters that includes 256 filters of $n \times 1$, 512 filters of $n \times 2$ and 256 filters of $n \times 3$. Here, $n$ represents the sequence's length. Such a combination can effectively extract multigranularity semantic information from the local. We set up three one-dimensional convolutions with filter sizes of 1, 2 and 3 instead of CNN blocks, and then compare the experimental results. The model is trained with different CNN layer in 50 epochs. In each experimental group, we save the model and test it using test data for every

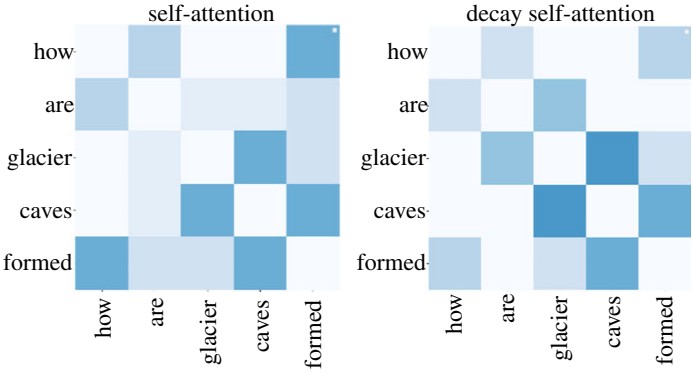

**Figure 9.** Attention weight of self-attention and decay self-attention.

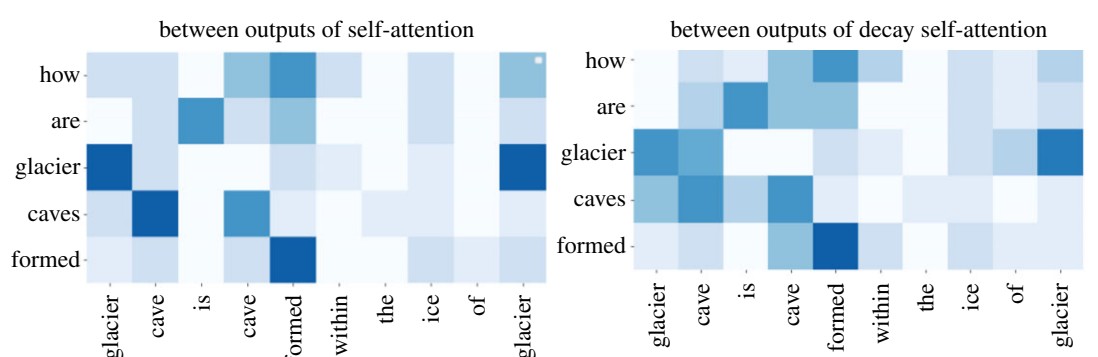

**Figure 10.** Attention weight of cross-attention between outputs of self-attention and between outputs of decay-attention.

**Table 3.** Result of different filters of CNN on TrecQA datasets.

| CNN layer | MAP | MRR |
|---|---|---|
| filters of size = 1 | 0.715 | 0.772 |
| filters of size = 2 | 0.779 | 0.842 |
| filters of size = 3 | 0.773 | 0.836 |
| filters of size = (1,2,3) | 0.818 | 0.879 |

epoch to obtain the highest MAP and MRR. As is shown in table 3, our CNN block yields better performance than the one-dimensional convolution of a single filter size.

## 5.2. Attention visualization

We use a pair of questions and correct answers in WikiQA to identify attention weights. Because our model uses multihead attention, the attention weight for each head differs; thus, the average of the attention weights of all heads is calculated. Figure 9 shows the attention weights of self-attention and decay self-attention in a question sentence. Decay self-attention is shown to have a more compact attention weight, and self-attention has more scattered attention weight. Decay self-attention has a larger weight between 'glacier' and 'caves' than self-attention. Perhaps the model is evaluating these words as a phrase, which is true. Decay self-attention also determines the relation between 'how' and 'formed', and has not forgotten them due to weight decaying. Thus, the two kinds of self-attention catch important words, but their focuses are slightly different.

Then, we use the same method to obtain the attention weight of cross-attention between outputs of self-attention and between outputs of decay-attention. As shown in figure 10, cross-attention between outputs of self-attention catches the same words, such as 'glacier', 'caves' and 'formed', which have large attention weights. The model is more likely to identify a question and answer that pay attention

**Table 4.** Results of different ablation models using the TrecQA dataset.

| model | MAP | MRR |
|---|---|---|
| DARCNN(full) | 0.818 | 0.879 |
| without BiLSTM | 0.751 | 0.809 |
| without BiLSTM (+ positional embedding) | 0.783 | 0.842 |
| without self-attention | 0.786 | 0.848 |
| without decay mask | 0.793 | 0.855 |
| without cross-attention | 0.801 | 0.860 |
| without CNN block | 0.760 | 0.818 |

to each other on the level of words. However, cross-attention between outputs of decay self-attention has a large attention weight for 'glacier' and 'caves', which makes the model more likely to make a question and answer pay attention to each other on the level of the phrase.

## 5.3. Ablation analysis

To demonstrate the effectiveness of the different components in our DARCNN model, an ablation experiment using the TrecQA dataset is designed. BiLSTM, self-attention, decay self-attention, cross-attention or a CNN block are removed from the original model. Similarly, each model was trained for 50 epochs and tested, with the highest MAP and MRR being recorded. We compared six ablation models and the result of DARCNN (full model) using the TrecQA dataset, as shown in table 4.

Without the BiLSTM layer, the MAP and MRR decreased by 6.7% and 7.0%, respectively; thus, BiLSTM has a large impact on the results. Without the BiLSTM layer, the data goes through the embedding layer and goes directly to self-attention. However, self-attention cannot obtain position and word order information in the sequence, which leads to the self-attention not generating the best attention weight. Therefore, the performance of the network model degrades. After adding positional embedding to the model to replace BiLSTM, the MAP and MRR only decreased by 3.5% and 3.7%. Therefore, we believe that BiLSTM considers the relationship between position and word order when generating the new representation matrix, which makes up for the lack of self-attention.

Without the self-attention layer, the MAP and MRR decreased by 3.2% and 3.1%, respectively. BiLSTM is limited by distance in global modelling, and self-attention mitigates this deficiency. From the experimental results, self-attention successfully determined the global features in the sentence.

Without the decay mask, decay self-attention becomes self-attention. In this case, the MAP and MRR decreased by 2.5% and 2.5%, respectively, which proves that decay self-attention and self-attention cannot replace each other. They respectively catch different granularities of information from the global and local perspectives.

Without the cross-attention layer, the MAP and MRR decreased by 1.7% and 1.9%, respectively. Other network components focus on semantic modelling of a single sequence, while cross-attention implements information interactions between two sequences. From the experimental results, cross-attention increases the performance of the network model.

Without the CNN block, the MAP and MRR decreased by 5.8% and 6.1%, respectively. Thus, the CNN block has a large impact on the results. These results show that the CNN yields good performance in answer selection and short text matching because the CNN can use filters to extract local semantic information of text sequences. By setting different filter sizes, semantic information of different granularities can be obtained.

Through ablation analysis, the different functions of each component to the model can be observed, and the previous discussion and analysis can be verified.

# 6. Conclusion

In this paper, a new deep model called DARCNN with two kinds of attention mechanism for answer selection is proposed. In DARCNN, self-attention can capture more feature information between words at any distance in a single sequence. Decay self-attention focuses more on local feature

information and is more suitable for answer selection with short text. We use cross-attention to capture interactive information in two sequences of questions and candidate answers. As we can see in the attention visualization, different information is obtained via cross-attention; thus, decay self-attention and self-attention focus on different features. The experimental results show that double attention can improve model performance to obtain better representation vectors of questions and answers.

Via experimentation and analysis, the BiLSTM and CNN block are also shown to have semantic modelling capabilities that are both global and local. They perform different functions and can make up for each other's shortcomings.

The DARCNN model surpassed other answer selection methods to achieve the most advanced performance when analysing four QA datasets. In future work, we will consider improving DARCNN and applying it to other NLP tasks, such as dialogue systems and reading comprehension.

Data accessibility. Data available from the Dryad Digital Repository: https://doi.org/10.5061/dryad.kkwh70s12 [36].
Authors' contributions. G.B. and Y.W. conceptualized this method; G.B. drafted the manuscript; X.S. and H.Z. validated the results; Y.W. revised the manuscript. All authors gave final approval for publication.
Competing interests. We have no competing interests.
Funding. This research was supported by National Natural Science Foundation of China (grant nos. 11802168, 61603238 and 51575331) and project funded by China Postdoctoral Science Foundation (grant no. 2019M661458).
Acknowledgements. The authors would like to thank the referees for their valuable suggestions.

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
