## [Reviewer comments · Royal Society Open Science]

Review History

RSOS-191517.R0 (Original submission)

Review form: Reviewer 1

Is the manuscript scientifically sound in its present form?

Yes

Are the interpretations and conclusions justified by the results?

No

Is the language acceptable?

Yes

Do you have any ethical concerns with this paper?

No

Have you any concerns about statistical analyses in this paper?

No

Recommendation?

Accept with minor revision (please list in comments)

Comments to the Author(s)

Technical concerns:

- [1] Page 5, line 49-50: this statement is likely to be faulty; the multiplication of V results in a new contextualized representation matrix for the inputs, instead of “attention weight”, which, in different contexts, refers to either the logit scores before softmax, or the distribution after softmax.
- [2] Eq (10): Does w here denote a vector? If so, how is Q^*K^T in Eq. (10) supposed to produce a vector? If not, you should not take w_i as a scalar in Eq. (11). In Eq. (14), you add w with your decay matrix, does that mean w is a matrix? These notations need to be fixed or further clarified.

General concerns:

It’s good to see the authors did an ablation study for all components in the framework. However, I still have the following concerns:

- [1] When you say “wo/ BiLSTM/attention/CNN”, did you use the same amount of parameters? For instance, if you want to prove the effects of decay attention, the best practice would be replacing it with another self-attention, i.e., just removing the decay mask. Completely removing this attention leads to a non-negligible loss in the parameter size, which most likely, hurts overall model capacity.
- [2] The comparison with BiLSTM is an interesting part to me, since the function of your decay attention seems similar to BiLSTM, i.e., putting emphasis on surrounding words. In order to demonstrate the point “self-attention cannot obtain the position information and word order information on the sequence”, I guess the authors should increase the layers of self-attention/decay attention when trying removing BiLSTM.
- [3] This paper argues that self-attention is not enough for capturing positional information. I generally agree with this point. However, in the original paper, together with the proposal of self-attention, “position embeddings” were adopted to encode positional prior of a sequence. I disappointedly found its reference/comparison was missing from this paper. Since the “decay matrix” proposed here is supposed to be one of the key contributions of this work, I would like to see a deeper analysis of this component, including the comparison to positional embeddings and more fair ablation study settings as mentioned above.

Review form: Reviewer 2

Is the manuscript scientifically sound in its present form?

No

Are the interpretations and conclusions justified by the results?

No

Is the language acceptable?

No

Do you have any ethical concerns with this paper?

No

Have you any concerns about statistical analyses in this paper?

Yes

Recommendation?

Major revision is needed (please make suggestions in comments)

Comments to the Author(s)

This manuscript presents a neural architecture for answer selection in an information retrieval system. The neural architecture consists of several components, including BiLSTM, self-, cross-

attention, CNN, and MLP layers. The proposed architecture outperforms some selected baseline methods.

While the manuscript presents an interesting problem and addresses relevant challenges, I believe the paper is not well-presented, and many choices of the authors are not motivated. Below I provide more details about the limitations of the paper.

While the authors refused to discuss the related work in detail (Section 3 is done very superficially), they have spent a lot of effort describing well-known theories and components of neural models (e.g., Sections 4.2, 4.3, etc.) Also, while providing details about these neural components, I didn't find the provided motivation of employing them convincing. Therefore, it is a question whether the choice of components is well-motivated, which will then affect the quality of the contribution of the manuscript.

Furthermore, important baselines are missing from the experiments. For instance, I am curious to know how their system will compare against a BERT-based model. Also, the depth of discussions and analyses is not convincing enough. Some recent collections for question answering are also missing (e.g., ANTIQUE [1]). I would like to recommend to the authors to redo their experiments also on this dataset.

Overall, I believe that this manuscript has the potential of making a good publication but requires extensive revision, including but not limited to improving the presentation (i.e., related work, model description, results and analysis). More recent baselines should be added (i.e., BERT-based models). More recent datasets that are more appropriate for testing neural models should also be added to the experiments (i.e., ANTIQUE).

[1] <https://ciir.cs.umass.edu/downloads/Antique/>

Decision letter (RSOS-191517.R0)

27-Feb-2020

Dear Dr Wei,

The editors assigned to your paper ("Double attention recurrent convolution neural network for answer selection") have now received comments from reviewers. We would like you to revise your paper in accordance with the referee and Associate Editor suggestions which can be found below (not including confidential reports to the Editor). Please note this decision does not guarantee eventual acceptance.

Please submit a copy of your revised paper before 21-Mar-2020. Please note that the revision deadline will expire at 00.00am on this date. If we do not hear from you within this time then it will be assumed that the paper has been withdrawn. In exceptional circumstances, extensions may be possible if agreed with the Editorial Office in advance. We do not allow multiple rounds of revision so we urge you to make every effort to fully address all of the comments at this stage. If deemed necessary by the Editors, your manuscript will be sent back to one or more of the original reviewers for assessment. If the original reviewers are not available, we may invite new reviewers.

To revise your manuscript, log into <http://mc.manuscriptcentral.com/rsos> and enter your Author Centre, where you will find your manuscript title listed under "Manuscripts with Decisions." Under "Actions," click on "Create a Revision." Your manuscript number has been

appended to denote a revision. Revise your manuscript and upload a new version through your Author Centre.

- Data accessibility

If you wish to submit your supporting data or code to Dryad (<http://datadryad.org/>), or modify your current submission to dryad, please use the following link:
<http://datadryad.org/submit?journalID=RSOS&manu=RSOS-191517>

- Competing interests

- Authors' contributions

- Acknowledgements

- Funding statement

on behalf of Professor Mirella Lapata (Associate Editor) and Marta Kwiatkowska (Subject Editor)
openscience@royalsociety.org

Associate Editor's comments (Professor Mirella Lapata):

Comments to the Author:

Dear Author,

the reviewers have read your manuscript and recommend various revisions that will make it stronger. While revising your paper, please answer the technical questions of reviewer 1 and also make sure to address the rewrites reviewer 2 recommends. Also in the experimental section, it would be good to see further analysis, e.g., on the decay matrix component and possibly the addition of a BERT-based baseline.

Reviewers' Comments to Author:

Reviewer: 1

Comments to the Author(s)

Technical concerns:

- [1] Page 5, line 49-50: this statement is likely to be faulty; the multiplication of V results in a new contextualized representation matrix for the inputs, instead of "attention weight", which, in different contexts, refers to either the logit scores before softmax, or the distribution after softmax.
- [2] Eq (10): Does w here denote a vector? If so, how is Q^*K^T in Eq. (10) supposed to produce a vector? If not, you should not take w_i as a scalar in Eq. (11). In Eq. (14), you add w with your decay matrix, does that mean w is a matrix? These notations need to be fixed or further clarified.

General concerns:

It's good to see the authors did an ablation study for all components in the framework. However, I still have the following concerns:

- [1] When you say "wo/ BiLSTM/attention/CNN", did you use the same amount of parameters? For instance, if you want to prove the effects of decay attention, the best practice would be replacing it with another self-attention, i.e., just removing the decay mask. Completely removing this attention leads to a non-negligible loss in the parameter size, which most likely, hurts overall model capacity.
- [2] The comparison with BiLSTM is an interesting part to me, since the function of your decay attention seems similar to BiLSTM, i.e., putting emphasis on surrounding words. In order to

demonstrate the point “self-attention cannot obtain the position information and word order information on the sequence”, I guess the authors should increase the layers of self-attention/decay attention when trying removing BiLSTM.

[3] This paper argues that self-attention is not enough for capturing positional information. I generally agree with this point. However, in the original paper, together with the proposal of self-attention, “position embeddings” were adopted to encode positional prior of a sequence. I disappointedly found its reference/comparison was missing from this paper. Since the “decay matrix” proposed here is supposed to be one of the key contributions of this work, I would like to see a deeper analysis of this component, including the comparison to positional embeddings and more fair ablation study settings as mentioned above.

Reviewer: 2

Comments to the Author(s)

This manuscript presents a neural architecture for answer selection in an information retrieval system. The neural architecture consists of several components, including BiLSTM, self-, cross-attention, CNN, and MLP layers. The proposed architecture outperforms some selected baseline methods.

While the manuscript presents an interesting problem and addresses relevant challenges, I believe the paper is not well-presented, and many choices of the authors are not motivated. Below I provide more details about the limitations of the paper.

While the authors refused to discuss the related work in detail (Section 3 is done very superficially), they have spent a lot of effort describing well-known theories and components of neural models (e.g., Sections 4.2, 4.3, etc.) Also, while providing details about these neural components, I didn't find the provided motivation of employing them convincing. Therefore, it is a question whether the choice of components is well-motivated, which will then affect the quality of the contribution of the manuscript.

Furthermore, important baselines are missing from the experiments. For instance, I am curious to know how their system will compare against a BERT-based model. Also, the depth of discussions and analyses is not convincing enough. Some recent collections for question answering are also missing (e.g., ANTIQUE [1]). I would like to recommend to the authors to redo their experiments also on this dataset.

Overall, I believe that this manuscript has the potential of making a good publication but requires extensive revision, including but not limited to improving the presentation (i.e., related work, model description, results and analysis). More recent baselines should be added (i.e., BERT-based models). More recent datasets that are more appropriate for testing neural models should also be added to the experiments (i.e., ANTIQUE).

[1] <https://ciir.cs.umass.edu/downloads/Antique/>

Author's Response to Decision Letter for (RSOS-191517.R0)

See Appendix A.

Decision letter (RSOS-191517.R1)

Dear Dr Wei:

On behalf of the Editors, I am pleased to inform you that your Manuscript RSOS-191517.R1 entitled "Double attention recurrent convolution neural network for answer selection" has been accepted for publication in Royal Society Open Science subject to minor revision in accordance with the referee suggestions. Please find the referees' comments at the end of this email.

The reviewers and Subject Editor have recommended publication, but also suggest some minor revisions to your manuscript. Therefore, I invite you to respond to the comments and revise your manuscript.

- Ethics statement

- Data accessibility

<http://datadryad.org/submit?journalID=RSOS&manu=RSOS-191517.R1>

- Competing interests

- Authors' contributions

- Acknowledgements

- Funding statement

Because the schedule for publication is very tight, it is a condition of publication that you submit the revised version of your manuscript before @@author due date will be populated when the email is sent@@. Please note that the revision deadline will expire at 00.00am on this date. If you do not think you will be able to meet this date please let me know immediately.

Kind regards,

Anita Kristiansen
Editorial Coordinator

on behalf of Professor Mirella Lapata (Associate Editor) and Marta Kwiatkowska (Subject Editor)
openscience@royalsociety.org

Editors' Comments to Author(s):

We recommend that you ask a native speaker of English or solicit the support of a language editing service (<https://royalsociety.org/journals/authors/language-polishing/>) prior to resubmitting the manuscript. Note also typo in 'Simaese Architecture' (should be Siamese).

Author's Response to Decision Letter for (RSOS-191517.R1)

See Appendix B.

Decision letter (RSOS-191517.R2)

Dear Dr Wei,

It is a pleasure to accept your manuscript entitled "Double attention recurrent convolution neural network for answer selection" in its current form for publication in Royal Society Open Science.

on behalf of Professor Mirella Lapata (Associate Editor) and Marta Kwiatkowska (Subject Editor)
openscience@royalsociety.org

Appendix A

Dear Editors and Reviewers,

We have carefully read all the comments, and the corresponding revisions have been carried out. We believe this will make this paper more professional and rigorous. I would like to express my gratitude to editor and reviewers for their efforts. We marked the changes in red (Responds to comments from reviewer 1) and green (Responds to comments from reviewer 2) in the revised manuscript. We hope it meets with approval.

The point to point responds are as follows:

Associate Editor's comments:

Comments to the Author:

Dear Author,

the reviewers have read your manuscript and recommend various revisions that will make it stronger. While revising your paper, please answer the technical questions of reviewer 1 and also make sure to address the rewrites reviewer 2 recommends. Also in the experimental section, it would be good to see further analysis, e.g., on the decay matrix component and possibly the addition of a BERT-based baseline.

Response: We have revised the manuscript according to the opinions of two reviewers and answered the reviewers' questions. In order to ensure the scientific nature of the article, we added some experiments. In the revised version we added an ablation experiment that only removed the decay matrix. This will not affect the capacity of the model, and the experimental result is that the MAP and MRR decreased by 2.5% and 2.5%, respectively. It can be seen that the decay mask has a positive effect on the model, and there was a clear difference between self-attention and decay self-attention. Then we listened to the suggestions of reviewers added BERT-based model and the data set of ANTIQUE to the experiment. Through experiments, compared with BERT, DARCNN is 1 to 2 percentage points lower on the three data sets, but the advantage of DARCNN is that its model capacity is much smaller than BERT's.

Reviewer: 1

Comments to the Author(s)

Technical concerns:

[1] Page 5, line 49-50: this statement is likely to be faulty; the multiplication of V results in a new contextualized representation matrix for the inputs, instead of "attention weight", which, in different contexts, refers to either the logit scores before softmax, or the distribution after softmax.

Response: In the original manuscript, the expression "multiply the matrix V to get a feature vector of the attention weight" was not accurate enough. I am willing to follow the suggestions of reviewers and change them to "multiply the matrix V to get a new contextualized representation matrix". Thank you very much for your suggestion.

[2] Eq (10): Does w here denote a vector? If so, how is $Q \cdot K^T$ in Eq. (10) supposed to produce a vector? If not, you should not take w_i as a scalar in Eq. (11). In Eq. (14), you add w with your decay matrix, does that mean w is a matrix? These notations need to be fixed or further clarified.

Response: In Eq. (10), w denotes a matrix. In the original manuscript, $W_i^Q \in \mathbb{R}^{d_{\text{model}} \times d_k}$, $W_i^K \in \mathbb{R}^{d_{\text{model}} \times d_k}$, so it's calculated that $Q \in \mathbb{R}^{d_n \times d_k}$, $K \in \mathbb{R}^{d_n \times d_k}$.

Then according to the Eq. (10), we can get $w \in \mathbb{R}^{d_{\text{model}} \times d_k}$. Eq. (11) is actually softmax. For clarity, in the revised version, Eq. (11) is directly expressed as softmax. Thanks.

General concerns:

It's good to see the authors did an ablation study for all components in the framework. However, I still have the following concerns:

[1] When you say "wo/ BiLSTM/attention/CNN", did you use the same amount of parameters? For instance, if you want to prove the effects of decay attention, the best practice would be replacing it with another self-attention, i.e., just removing the decay mask. Completely removing this attention leads to a non-negligible loss in the parameter size, which most likely, hurts overall model capacity.

Response: I am willing to follow the comments of reviewers. In order to better prove the role of decay self-attention in the model, I only removed the decay mask in the ablation experiment for comparison. From the results of the experiment, the MAP and MRR decreased by 2.5% and 2.5%, respectively. Thank you.

[2] The comparison with BiLSTM is an interesting part to me, since the function of your decay attention seems similar to BiLSTM, i.e., putting emphasis on surrounding words. In order to demonstrate the point "self-attention cannot obtain the position information and word order information on the sequence", I guess the authors should increase the layers of self-attention/decay attention when trying removing BiLSTM.

Response: The point "self-attention cannot obtain the position information and word order information on the sequence" is obtained according to other literature and the calculation principle of self-attention. In the ablation experiment to remove BiLSTM, the MAP and MRR decreased by 6.7% and 7.0%, respectively. In the modified version, we used positional embedding to replace BiLSTM, the MAP and MRR only decreased by 3.5% and 3.7%. Therefore, we believe that BiLSTM considers the relationship between positional information and word order, while self-attention does not.

[3] This paper argues that self-attention is not enough for capturing positional information. I generally agree with this point. However, in the original paper, together with the proposal of self-attention, “position embeddings” were adopted to encode positional prior of a sequence. I disappointedly found its reference/comparison was missing from this paper. Since the “decay matrix” proposed here is supposed to be one of the key contributions of this work, I would like to see a deeper analysis of this component, including the comparison to positional embeddings and more fair ablation study settings as mentioned above.

Response: The location information and word order information that BiLSTM has considered. I am willing to listen to the comments of reviewers and compare with positional embedding. In the ablation experiment of removing BiLSTM, we used positional embedding to replace BiLSTM. The MAP and MRR only decreased by 3.5% and 3.7%. Therefore, we believe that BiLSTM takes into account the relationship between position and word order when generating the new representation matrix, which makes up for the lack of self-attention. Thank you.

Reviewer: 2

Comments to the Author(s)

This manuscript presents a neural architecture for answer selection in an information retrieval system. The neural architecture consists of several components, including BiLSTM, self-, cross-attention, CNN, and MLP layers. The proposed architecture outperforms some selected baseline methods.

While the manuscript presents an interesting problem and addresses relevant challenges, I believe the paper is not well-presented, and many choices of the authors are not motivated. Below I provide more details about the limitations of the paper.

While the authors refused to discuss the related work in detail (Section 3 is done very superficially), they have spent a lot of effort describing well-known theories and components of neural models (e.g., Sections 4.2, 4.3, etc.) Also, while providing details about these neural components, I didn't find the provided motivation of employing them convincing. Therefore, it is a question whether the choice of components is well-motivated, which will then affect the quality of the contribution of the manuscript.

Furthermore, important baselines are missing from the experiments. For instance, I am curious to know how their system will compare against a BERT-based model. Also, the depth of discussions and analyses is not convincing enough. Some recent collections for question answering are also missing (e.g., ANTIQUE [1]). I would like to recommend to the authors to redo their experiments also on this dataset.

Overall, I believe that this manuscript has the potential of making a good publication but requires extensive revision, including but not limited to improving the presentation (i.e., related work, model description, results and analysis). More recent baselines should be added (i.e., BERT-based models). More recent datasets that are more appropriate for testing neural models should also be added to the experiments (i.e., ANTIQUE).

Response: First, the motivation of designing this model is to improve the existing self-attention and make the model better in the answer selection. Secondly, the flexibility of self-attention feature extraction makes it easy to improve. In this article, we proposed adding a decay mask on self-attention, called decay self-attention. And use cross-attention to merge the two kinds of self-attention. To further demonstrate the role of these two types of self-attention in the model, in the modified version we added an ablation experiment that only removed the decay matrix. This will not affect the capacity of the model, and the experimental result is that the MAP and MRR decreased by 2.5% and 2.5%, respectively. It can be seen that the decay mask has a positive effect on the model, and there was a clear difference between self-attention and decay self-attention. BiLSTM is also an important component of the model. In the modified version, we added an ablation experiment and replaced BiLSTM with positional embedding. Through experimental comparison, we believe that BiLSTM can make up for the lack of location information and word order relation obtained by self-attention.

Then we listened to the suggestions of reviewers added BERT-based model and the data set of ANTIQUE to the experiment. Through experiments, compared with BERT, DARCNN is 1 to 2 percentage points lower on the three data sets, but the advantage of DARCNN is that its model capacity is much smaller than BERT's. Thank you very much for your suggestion.

Appendix B

Dear Editors and Reviewers,

We have carefully read all the comments, and the corresponding revisions have been carried out. We believe this will make this paper more professional and rigorous. I would like to express my gratitude to editor and reviewers for their efforts. We hope it meets with approval.

The point to point responds are as follows:

Editor Comments to Author:

We are marking this as 'accept with minor' to allow for improvements in English. Note also typo in 'Simaese Architecture' (should be Siamese).

As you have been requested to edit the written English, you must provide proof that you have done so: acceptable proof includes a certificate of language-editing from a language editing service or a signed letter from a native speaker of English. If you do not provide this proof, your manuscript may be returned to you.

For information about language editing services endorsed by the Royal Society, please follow the link below:

<https://royalsociety.org/journals/authors/language-polishing/>

Response: We have revised the typo in the manuscript.

We have done the language editing service suggested by the editor, and get a certificate of language-editing. This proof has been uploaded in the submission system. Thank you very much for your suggestion.